# Genetic Structure and Phylogeography of Commercial *Mytilus unguiculatus* in China Based on Mitochondrial *COI* and *Cytb* Sequences

**Xuelian Wei, Zeqin Fu, Jiji Li, Baoying Guo and Yingying Ye \***

National Engineering Research Center for Marine Aquaculture, Zhejiang Ocean University, Zhoushan 316022, China

\* Correspondence: yeyy@zjou.edu.cn

**Abstract:** In order to study the genetic structure and population geographic distribution pattern of coastal mussel populations in the coast of China, mitochondrial DNA (*COI* and *Cytb* genes) were used to analyze the genetic diversity, genetic structure, and population history dynamics of *Mytilus unguiculatus* in the East China Sea and the Yellow Sea. We detected high levels of genetic diversity in seven populations of *M. unguiculatus*. A total of 34 haplotypes of *COI* genes and 29 haplotypes of *Cytb* were obtained. The haplotype diversity of *COI* ranged from around 0.77 to 0.93 (*Cytb*: 0.83~0.91). The nucleotide diversity of *COI* ranged from around 0.0044 to 0.0064 (*Cytb*: 0.0049~0.0063). The coefficient of genetic differentiation ($F_{ST}$) of *COI* ranged from around 0.031 to 0.080, and *Cytb* ranged from around 0.028 to 0.039. Analysis of molecular variance (AMOVA) and a phylogenetic tree showed that the genetic structure was relatively weak, and there was no clear population differentiation. The neutrality test results showed that Tajima's D value and Fu's *Fs* value were not significant, and no significant population demographic events, including population expansion or population bottleneck, were detected in the samples. The Bayesian skyline graph analysis showed that the effective population size has been relatively stable for nearly 10,000 years, without any large population fluctuations. It was speculated that the seven populations in the present study should belong to the same group. This study provides a comprehensive survey of the genetic characteristics of *M. unguiculatus*, filling the gaps among related studies. It provides theoretical support and material accumulation for seed selection and breeding, genetic resources' protection, and breeding management in the future.

**Keywords:** *Mytilus unguiculatus*; mitochondrial DNA; genetic diversity; genetic structure; phylogeography

## 1. Introduction

The hard-shelled mussel has different scientific names, such as *Mytilus coruscus* and *Mytilus unguiculatus*. Although "*Mytilus coruscus*" is widely used in the bibliography, this form is not accepted according to the World Register of Marine Species (WoRMS; http://www.marinespecies.org; accessed on 14 January 2023). Therefore, this study uses "*Mytilus unguiculatus*" as the scientific name of this mussel. This species inhabits the temperate zone along the coastal water of the Indo-Pacific region. According to the Food and Agriculture Organization (FAO), global capture production of *M. unguiculatus* was rapidly expanding during the 2000s, with a peak of 3399 tons in 2009, followed by a sustained decline (633 tons in 2020) [1]. This may be due to the immature breeding technology, the damage to the aquaculture area and the overfishing of *M. unguiculatus* [2]. With a unique flavor and delicious taste, *M. unguiculatus* is one of the most popular seafoods among Chinese consumers. Plenty of aquaculture practitioners invest in mussel farming. Therefore, this species is an economically important species in China, and undergoes large-scale breeding in the East China Sea [3]. The dried mussel has been used as a medicine in traditional Chinese therapy as well as in modern medicine [4,5]. Many practitioners

suggest that patients frequently consume mussel to regulate liver and kidney function, boost the immune system, and so on. Despite that this species is rich in physiological and biochemical functions and disease control, genetic information is still lacking. Although in recent years different genome assemblies of *M. unguiculatus* have been published [6,7], population genetics/genomics studies are still scarce. There is no large-scale population genetic research, most of which are limited to a certain area. Small-scale population genetic research cannot well-reflect the current status of genetic resources of mussels in coastal areas of this country and is not conducive to understanding their true genetic structure and population geographic distribution.

The genetic material of the mitochondria is a hotspot for genetic analysis. As molecular markers, the highest conserved genes are the first choice, such as the cytochrome c oxidase subunits I–III (*COI-III*), cytochrome b (*Cytb*), and control (D-loop) regions [8–10]. They have the characteristics of a simple molecular structure, strict maternal inheritance, no recombination, no common sequence with the nuclear genome, fast evolution rate, and multiple copies [11–15]. Compared with other genes in mtDNA, these gene regions can provide enough sequence information to search for suitable genetic variation and are amenable to amplification and sequencing.

In this study, based on the basic theory of molecular phylogeography and classical research cases, the genetic structure and population geographical distribution pattern of *M. unguiculatus* in China's coastal areas were studied by using mitochondrial DNA molecular markers, especially the main producing areas of *M. unguiculatus* in Zhejiang and Fujian coastal areas. Through the genetics analyses between different local populations, we attempt to provide insights into the genetic variation and historical dynamics of *M. unguiculatus* in the main producing areas (Zhejiang and Fujian Provinces). We think that this study can provide suggestions for the design of a comprehensive strategy for protecting genetic resources.

## 2. Materials and Methods

### 2.1. Sample Collection

The samples of *M. unguiculatus* were collected from the following seven collection sites: Qingdao (QD), Zhoushan (ZS), Xiangshan (XS), Yuhuan (YH), Taishan (TS), Pingtan (PT), and Xiamen (XM), in the China Sea (Figure 1; Table 1). Fragments of adductor muscle of all individuals were collected from each mussel and stored in 95% ethanol under −20 °C until DNA extraction.

**Table 1.** The sampling details of the seven sampling locations.

| Sampling Site | Abbreviation | Coordinate | Sampling Date | Sample Size | |
|---|---|---|---|---|---|
| | | | | *COI* | *Cytb* |
| Qingdao | QD | 35°55′ N, 120°30′ E | September 2016 | 29 | 31 |
| Zhoushan | ZS | 30°12′ N, 122°42′ E | January 2017 | 27 | 26 |
| Xiangshan | XS | 29°14′ N, 121°58′ E | December 2016 | 23 | 30 |
| Yuhuan | YH | 28°14′ N, 121°24′ E | March 2017 | 29 | 29 |
| Taishan | TS | 26°57′ N, 120°47′ E | November 2016 | 25 | 22 |
| Pingtan | PT | 25°34′ N, 119°47′ E | February 2017 | 29 | 29 |
| Xiamen | XM | 24°35′ N, 118°23′ E | November 2016 | 27 | 28 |
| Total | | | | 189 | 195 |

### 2.2. DNA Extraction and Polymerase Chain Reaction Amplification

The genomic DNA of *M. unguiculatus* was extracted by the salt-extraction method [16]. DNA was stored at − 20 °C for further analysis. Using the Primer premier v6.0 [17], primers were designed based on the complete mitochondrial genome in the NCBI database (GenBank accession: KJ577549). PCR amplification was performed using the following primer pairs: coi-F, 5′-GCACTAAGTGAGTCCTTGTTAAAT-3′; coi-R, 5′-ATTGATAGCCAAATCA AATTGTAAC-3′; cytb-F, 5′-TGAGCTGTAAACTCATGAA- CAAG-3′, cytb-R, 5′-TTAGTAA

GAAGAATCACGCATCC-3′. PCR amplification was performed in 25 μL volumes, containing 12.5 μL of CW0716 2 × Taq MasterMix (Cwbiotech. Co. Ltd., Peking, China), 1 μL of each forward and reverse primers, 0.5 μL of template DNA, and 10 μL of ultrapure water. PCR amplification was performed using an Applied Biosystems Veriti 96-Well Thermal Cycler (Applied Biosystems, Inc., Foster City, CA, USA). After initial denaturation at 95 °C for 3 min, reactions were cycled for 30 cycles at 95 °C for 30 s, 51 °C (*Cytb*: 52 °C) for 30 s (each pair of primers was screened in gradient temperature), 72 °C for 1.5 min, and final elongation at 72 °C for 10 min. The products were electrophoresed in 1% agarose. Successful reactions were sent to a commercial facility for sequencing (Tsingke Biotehcnology Co., Ltd., Hangzhou, China).

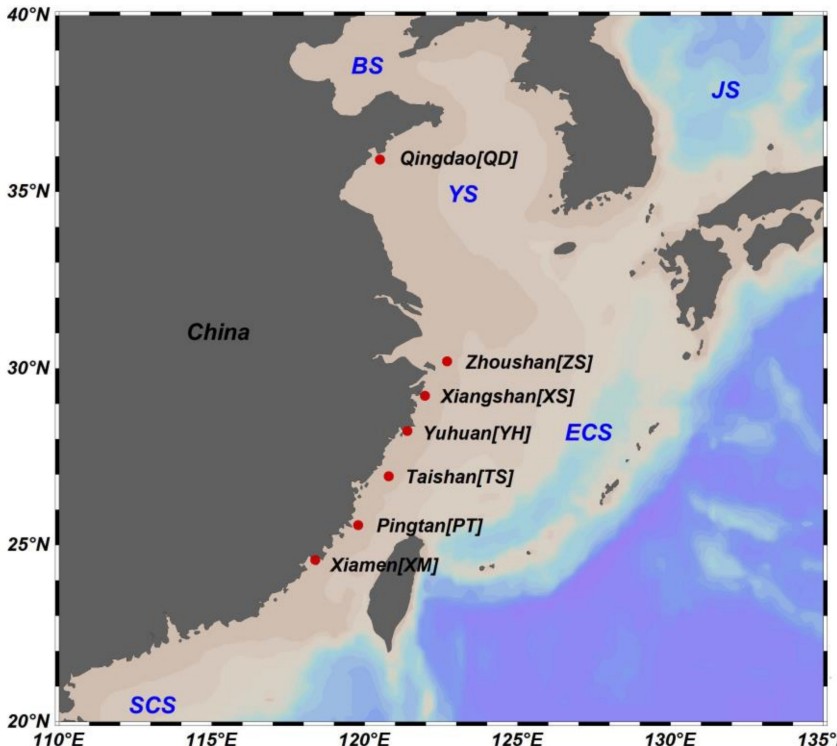

**Figure 1.** The map of sampling locations (BS: Bohai Sea; YS: Yellow Sea; ECS: East China Sea; SCS: South China Sea; JS: Japan Sea).

*2.3. Genetic Data Analysis*

BioEdit v7.25 [18] was used to analyze the sequences. Using Seqman v8.1 [19], the sequences from different regions were concatenated. The program Clustal X v2.0 [20] was used to create a standard FASTA file and submitted to BLAST tools in NCBI (https://blast.ncbi.nlm.nih.gov/Blast.cgi; accessed on 25 October 2022) for blasting, which ensured that the analyzed individuals were from *M. unguiculatus*.

MEGA v7.0 software [21] was used to align the fragments. The program jModelTest v2.0 [22] was performed to test the best nucleotide substitution model. Using DnaSP v6.0 [23], basic population genetic parameters, such as the number of haplotypes (n), haplotype diversity (h), and nucleotide diversity (π), were obtained. An analysis of molecular variance (AMOVA) was performed on the seven sampling locations [24]. Arlequin v3.5 [25] was used to calculate the pairwise $F_{ST}$ values (with 10,000 permutations, GTR model, and 0.5 Gamma rate) and the multiple simultaneous tests of $F_{ST}$ values were adjusted using the sequential Bonferroni procedure [26]. Arlequin v3.5 was also used to perform the AMOVA, evaluate the $\Phi$ statistic, and carry out neutrality tests (Tajima's D value and Fu's *Fs* values, with 10,000 permutations).



The program package Phylip v3.6 [27] was used to construct neighbor-joining (*NJ*) phylogenetic trees [28] based on all haplotypes with 1000 bootstraps. The program MrBayes v2.0 [29] was utilized to construct Bayesian phylogenetic trees based on all haplotypes. Then, the R package ggtree [30] was performed to merge the two kinds of phylogenetic trees. The software Network v5.1 [31] was used to plot the haplotypes network based on the median-joining method [32]. Using the program BEAST v2.0 [33], the historical dynamic was analyzed based on the Bayesian method. The program Tracer v1.4 [34] was used to construct the Bayes skyline plot (BSP), with 100,000,000 permutations, sampling frequency = 1000, and deleting the 10% aging value. The evolutionary rate was set at 2.4% per million years based on the evolutionary rate of bivalve reported by Hellberg and Vacquier [35] and Wood et al. [36].

## 3. Results

### 3.1. Genetic Diversity among Samples

In this study, alignment of 1656 bp of the *COI* gene was analyzed in 189 individuals and uncovered a total of 34 haplotypes (GenBank accession: MG214410-MG214443). In each of the seven samples (Table 2), the sample sizes (*N*) ranged from 23 (XS) to 29 (QD, YH, and TS). The number of haplotypes (*n*) observed ranged between 9 (QD) and 17 (YH). The haplotype (*h*) and nucleotide diversity (*π*) ranged from 0.77 (QD) to 0.93 (ZS), and 0.0044 (QD) to 0.0064 (ZS), respectively. With 54 individuals, MC-*COI*-Hap_02 was the most common central haplotype, accounting for 28.6%. The distribution of haplotypes was dispersed and a significant genetic divergence was not found (Figure 2a).

**Table 2.** The samples' details and the mitochondrion DNA diversity indices for *M. unguiculatus*.

| Sample | *COI* | | | | *Cytb* | | | |
|---|---|---|---|---|---|---|---|---|
| | *N* | *n* | *h* | *π* | *N* | *n* | *h* | *π* |
| QD | 29 | 9 | 0.77 ± 0.06 | 0.0044 ± 0.0007 | 31 | 12 | 0.83 ± 0.06 | 0.0052 ± 0.0006 |
| ZS | 27 | 15 | 0.88 ± 0.05 | 0.0059 ± 0.0004 | 26 | 11 | 0.88 ± 0.05 | 0.0063 ± 0.0006 |
| XS | 23 | 10 | 0.93 ± 0.03 | 0.0064 ± 0.0003 | 30 | 12 | 0.89 ± 0.04 | 0.0063 ± 0.0003 |
| YH | 29 | 17 | 0.88 ± 0.06 | 0.0054 ± 0.0007 | 29 | 11 | 0.84 ± 0.06 | 0.0058 ± 0.0006 |
| PT | 25 | 15 | 0.88 ± 0.03 | 0.0058 ± 0.0005 | 22 | 11 | 0.84 ± 0.06 | 0.0055 ± 0.0007 |
| TS | 29 | 11 | 0.92 ± 0.04 | 0.0047 ± 0.0007 | 29 | 11 | 0.90 ± 0.04 | 0.0049 ± 0.0008 |
| XM | 27 | 10 | 0.86 ± 0.04 | 0.0055 ± 0.0004 | 28 | 14 | 0.91 ± 0.03 | 0.0066 ± 0.0006 |
| Total | 189 | 34 | 0.88 ± 0.02 | 0.0054 ± 0.0002 | 195 | 29 | 0.86 ± 0.02 | 0.0058 ± 0.0002 |

Notes: *N*: sample size; *n*: number of haplotypes; *h*: haplotype diversity; *π*: nucleotide diversity.

Alignment of 1308 bp of the *Cytb* gene was analyzed with 195 individuals, obtaining 29 unique haplotypes (GenBank accession: MG21444–MG214472). Among the samples (Table 2), the sample sizes (*N*) ranged from 22 (PT) to 31 (QD). The number of haplotypes (*n*) observed ranged between 11 (except QD and XS) and 14 (XM). The haplotype diversity (*h*) ranged from 0.83 (QD) to 0.91 (XM). The nucleotide diversity (*π*) was between 0.0049 (QD) and 0.0063 (ZS, XS). The MC-*Cytb*-Hap_03 was the most common central haplotype, accounting for 31.8%. The distribution of haplotypes was also dispersed and a significant genetic divergence was not found (Figure 2b).

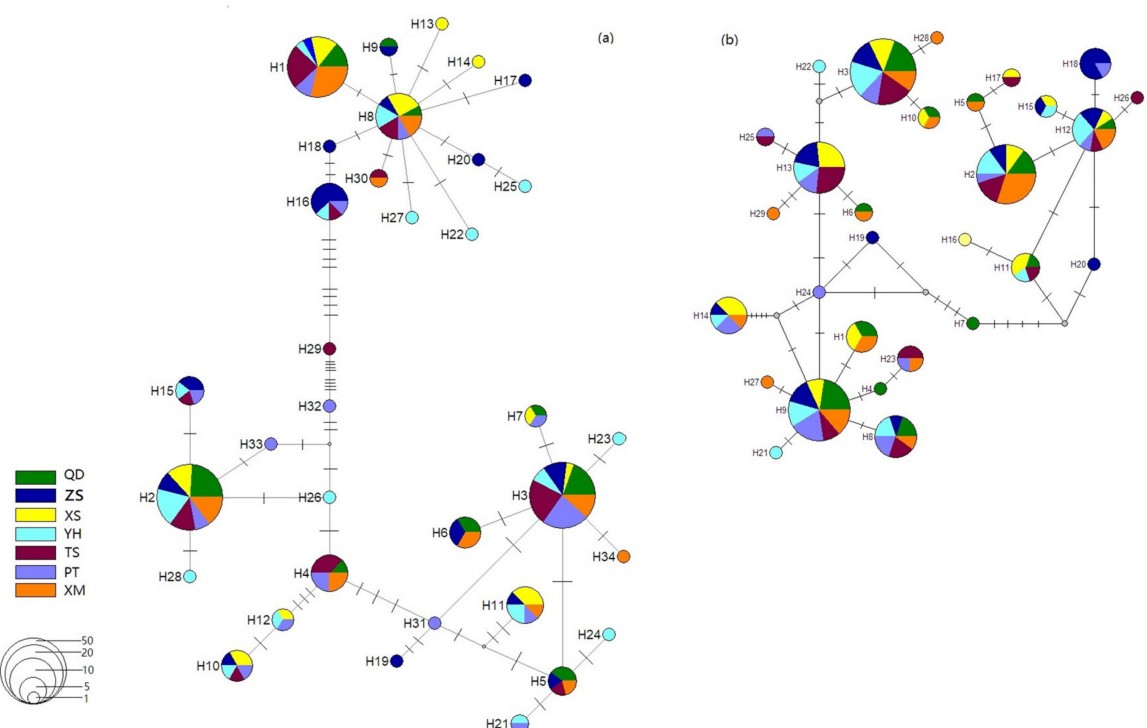

**Figure 2.** Haplotype network of *M. unguiculatus* from seven samples. (**a**) The network of *COI* genes. (**b**) The network of *Cytb* genes.

### 3.2. Genetic Divergence among Samples

The pairwise $F_{ST}$ analysis showed no significant divergence among samples for both mtDNA markers (*COI*:−0.031 to 0.080, *Cytb*:−0.028 to 0.039; Table 3). All *p*-values of the pairwise $F_{ST}$ indicated no statistically significant difference from zero (*p* > 0.002, after Bonferroni correction = 0.05/21). The AMOVA (Table 4) showed that the percentage of variation among samples was 0.1% (*Cytb*: 0.6%), and 98.2% (*Cytb*: 99.3%) within samples. Considering the $\Phi$ statistics, $\Phi_{SC}$ = 0.001 (*Cytb*: 0.001) and $\Phi_{ST}$ = 0.016 (*Cytb*: 0.006). For the phylogenetic trees, each marker produced two clusters. However, the haplotypes in trees did not cluster based on their geographic position. They were dispersed similarly to the results of the haplotype network (Figure 3).

**Table 3.** The pairwise $F_{ST}$ between the seven samples of *M. unguiculatus* based on *COI* (below diagonal) and *Cytb* gene (above diagonal).

| Sample | QD | ZS | XS | YH | PT | TS | XM |
|---|---|---|---|---|---|---|---|
| QD | - | 0.022 | 0.005 | −0.020 | −0.016 | −0.024 | −0.004 |
| ZS | 0.036 | - | −0.012 | −0.010 | 0.039 | 0.011 | −0.020 |
| XS | 0.080 | −0.001 | - | −0.027 | 0.013 | −0.015 | −0.016 |
| YH | −0.015 | −0.013 | 0.034 | - | 0.003 | −0.028 | −0.018 |
| PT | −0.021 | 0.016 | 0.053 | −0.016 | - | −0.007 | 0.017 |
| TS | 0.014 | −0.026 | −0.013 | −0.014 | −0.003 | - | −0.005 |
| XM | 0.017 | −0.024 | −0.006 | −0.004 | 0.010 | −0.031 | - |

**Table 4.** The AMOVA of the two mitochondrial markers between the seven samples of *M. unguiculatus*.

| Marker | Source of Variation | Degrees of Freedom | Sum of Squares | Φ Statistic | Percentage of Variation | *p*-Value |
|---|---|---|---|---|---|---|
| *COI* | Among samples | 6 | 21.7 | $\Phi_{SC} = 0.001$ | 0.1 | 0.032 |
| | Within samples | 182 | 809.4 | $\Phi_{ST} = 0.016$ | 98.2 | 0.031 |
| *Cytb* | Among samples | 6 | 15.3 | $\Phi_{SC} = 0.001$ | 0.6 | 0.050 |
| | Within samples | 188 | 684.8 | $\Phi_{ST} = 0.006$ | 99.3 | 0.023 |

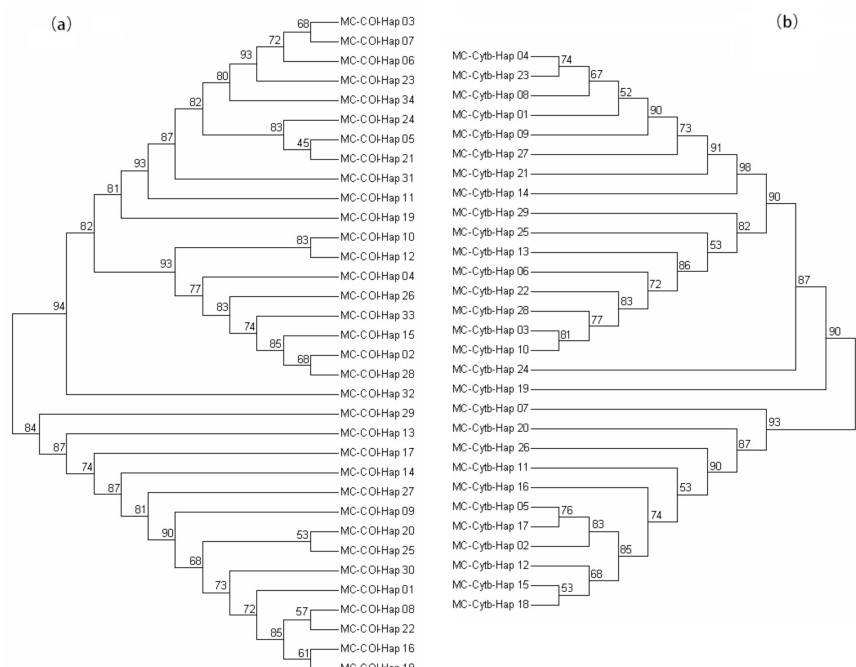

**Figure 3.** The phylogenetic tree of the seven samples of *M. unguiculatus* based on two markers. (**a**) The tree based on *COI*. (**b**) The tree based on *Cytb* (the Bayesian clade posterior probabilities (>0.50), the *NJ* bootstrap support values (>50%)).

### 3.3. Historical Dynamics

In the neutrality tests (Table 5), the *p*-values of Tajima's D and Fu's *Fs* did not support that the seven populations experienced clear demographic events based on the analysis of *COI* and *Cytb*. The generation time was one year based on the sexual maturity time of *M. unguiculatus*. Figure 4 revealed that the evolution of the effective population size was stable, as confirmed by the results of the neutrality test.

**Table 5.** Results of neutrality tests in the seven samples of *M. unguiculatus* for the two genetic markers employed.

| Sample | *COI* | | *Cytb* | |
|---|---|---|---|---|
| | Tajima's D | Fu's *Fs* | Tajima's D | Fu's *Fs* |
| QD | 1.42 (0.91) | 2.63 (0.91) | 0.93 (0.81) | 0.26 (0.60) |
| ZS | 2.13 (0.96) | 2.01 (0.90) | 1.32 (0.90) | 0.86 (0.72) |
| XS | 2.04 (0.93) | −0.64 (0.55) | 1.34 (0.88) | 1.07 (0.74) |
| YH | 1.02 (0.79) | −2.33 (0.24) | 1.25 (0.89) | 1.08 (0.74) |
| PT | 0.56 (0.66) | −2.25 (0.20) | −0.36 (0.43) | −0.66 (0.41) |
| TS | 1.74 (0.92) | 2.23 (0.92) | 0.82 (0.80) | 0.99 (0.71) |
| XM | 2.62 (0.92) | 2.51 (0.93) | 0.83 (0.77) | −0.68 (0.46) |

Notes: in brackets is the *p*-value.

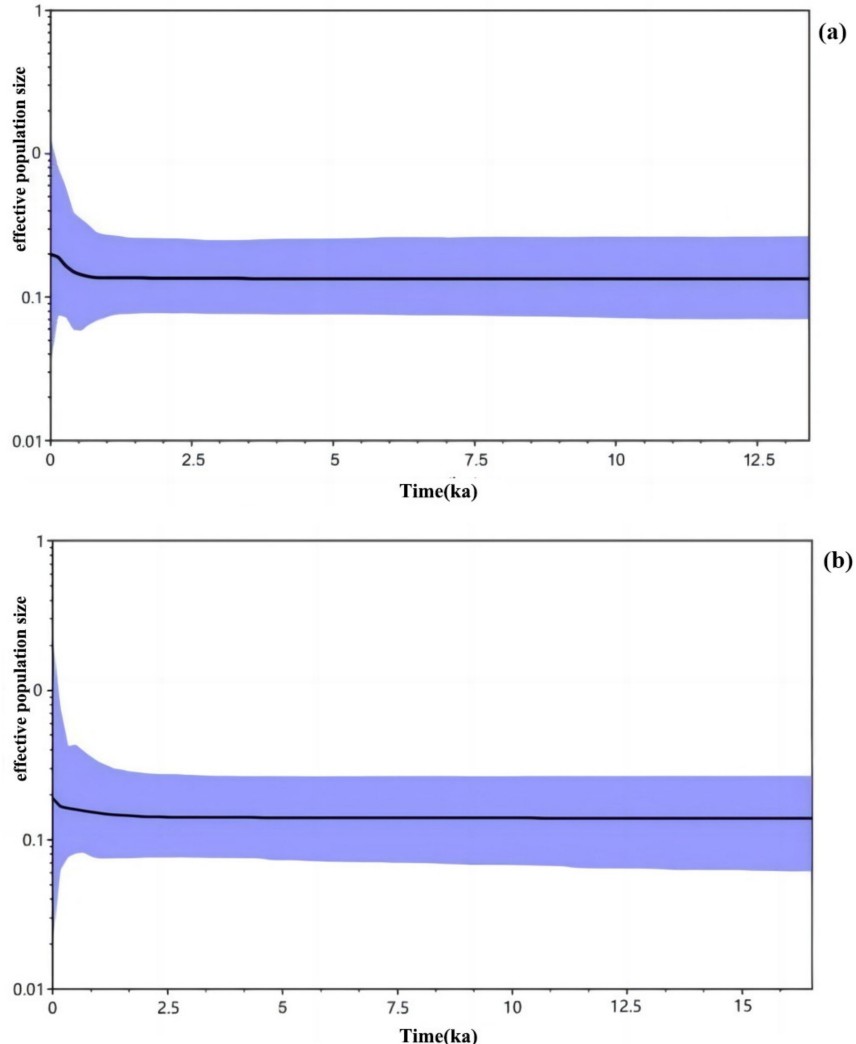

**Figure 4.** The BSP analysis of mitochondrial genes. (**a**) *COI* gene. (**b**) *Cytb* gene (black line is the median value, blue is the range of 95% CI).

### 4. Discussion

#### 4.1. Genetic Diversity of M. unguiculatus

Haplotype and nucleic acid diversity are two important indicators for evaluating population genetic diversity [37,38]. In the seven populations of *M. unguiculatus*, mtDNA markers were used to evaluate the genetic diversity. The results showed that the genetic diversity of all samples was high, as revealed by haplotype diversity (*COI*: 0.77~0.93, *Cytb*: 0.83~0.91) and nucleotide diversity (*COI*: 0.0044~0.0064, *Cytb*: 0.0049~0.0063). This result confirmed reports of the same species in the same areas by Guan et al. [39], Ye et al. [40], and Yang et al. [41]. Genetic diversity is an important part of biodiversity, and its level reflects the environmental adaptability, evolutionary potential, and viability of species. The higher the level of genetic diversity, the stronger the ability of the species to adapt to the environment, and the higher the evolutionary potential and the survival ability [39,42]. The high genetic diversity of mussels would indicate that *M. unguiculatus* resources in Chinese coastal areas were protected in the short term and could have good development prospects.

#### 4.2. Genetic Structure of M. unguiculatus

The genetic differentiation index can indirectly reflect the genetic differentiation status and the level of gene exchange among populations. When $0 < F_{ST} < 0.05$, there was no significant differentiation between groups, and when $0.5 < F_{ST} < 0.15$, the degree

of differentiation between groups was moderate [43–45]. In the present study, genetic divergence among samples of *M. unguiculatus* was mostly not significant, and it was confirmed by the results of haplotype networks. The AMOVA provided evidence that most genetic variation of *M. unguiculatus* came from within samples. The genetic variation of *M. unguiculatus* among samples was lower, which could explain why there was no significant genetic divergence. A phylogenetic tree of haplotypes showed two main clusters, but it was shown that the evolutionary relationship of the samples did not conform to the geographic distribution between samples. The same results were also found in studies of *Cyclina sinensis* [46] and *Chlamys Farreri* [47]. It is possible that the genetic structure of *M. unguiculatus* is weak and single, the reasons for which are outlined below.

In general, species with longer planktonic periods are able to drift for longer distances, driven by ocean currents [48]. Shanks et al. [49] found that the mollusc *Haliotis rubra*, with a 6-day pelagic larval phase, had a dispersal distance of less than 15 km, and that *Perna perna*, with a 15- to 20-day pelagic larval phase, could disperse to about 235 km. Species with a short planktonic stage drifted closer, and gene exchange was hindered, and their distribution patterns were more likely to produce complex lineage-geographic structures. For *M. unguiculatus*, its lifecycle had a long planktonic larval stage, about 3–4 weeks [50]. In this period, the larvae were unable to swim in the environment but passively followed the currents. At the end of the larval stage, the larvae become the juvenile mussel and start a fixed life. The larvae spreading in the ocean were similar to the gene flow among samples. With a long larval stage, it was possible for larvae to spread far away from the birthplace and go to other population habitats, which would cause the genetic exchange. The duration of gene flow coincided with the larval stage.

For analyzing the environment, in this study, the analyzed species was mainly distributed in the southeast coastline of China. Its habitats were reported from the Fujian province in China to Japan. In the wide marine environment, the ocean currents system was very complex [51]. In different seasons, the flow direction of ocean currents was different. In the East China Sea (ECS), the spawning of *M. unguiculatus* was in winter. The coastal currents flowed from north to south and the rivers were in the dry season, which means less fresh water flowed into the sea through the estuary. The less diluted water could not make an impact on coastal currents. The enhancement of coastal currents also had a circulation impact within the sea area [52]. Therefore, the results of genetic divergence among samples of *M. unguiculatus* confirmed the direction of the larvae following the ocean currents. The currents could be a factor to decrease the genetic variation among samples.

As an economical marine-cultured mussel, *M. unguiculatus* has a wide culture market. With the development of artificial breeding, practitioners would buy the seedling, and then transport the seed to breed in a suitable place. However, the development of the breeding method was backward. The use of open ponds in a natural environment resulted in the loss of larvae in the spawning period. The cultured larvae were mixed with the natural larvae and spread in the ocean. It was notable that the parents of cultured larvae were from a single source. The mixture decreases the genetic diversity and genetic variation among samples. At present, the development of artificial seeding is at a standstill. Single seedling selection resources have also resulted in the inbreeding of breeding samples. Without rational management of aquaculture and natural resources' protection, large-cultured larvae easily dilute the genetic diversity and genetic variation of natural larvae. This could be a factor of weak genetic structure in *M. unguiculatus*.

### 4.3. Protection and Breeding Strategies

In conclusion, from all the results of this study, the breeding strategies need to consider several aspects, as follows.

For species protection, the awareness of species protection should be enhanced. It is incomprehensive to measure the importance of species protection by the population size of species. The sustainable development of economic species requires a certain scale of genetic variation information. It can help to keep an excellent germplasm resource stock. This

is crucial for the further selection and breeding. The sustainable development of species resources can ensure the sustainable operation and output of the industry.

In mussel aquaculture, most practitioners tend to use two or more high-yield varieties with high genetic diversity. This has a large impact for other genotypes within species. In a length term of artificial seeding, this pattern makes it easy to lose rare variants. Without distinguishing between the local and foreign breeds, the artificial seeding can quickly alter the local genetic integrity of populations [53]. These factors will result in a decline in germplasm resources' quality. The government should strengthen the protection of native species, which is beneficial to the protection of the local species and helpful for them to evolve with other species under the same ecological environment.

In recent studies, the decline of *M. unguiculatus* resources has been reported [54,55]. Therefore, setting up a corresponding genetic protection unit may represent an effective way to protect germplasm genetic resources. It should also limit the physical communication between cultured populations and the wild ones to reduce the possibility of gene flow between them as much as possible. The existence of this gene flow was confirmed by Guan et al. [39] and Shen et al. [56]. Furthermore, these wild populations represent aquaculture stocks' primary source of genetic variability [56]. Hybridization and genetic introgression also exist among different mussel species (e.g., *Mytilus edulis* [57,58]). *M. unguiculatus* mainly anchors in the shallow sea area below the low-tide line [59], with many overlapped breeding sites in the Zhejiang and Fujian Provinces of China [60,61]. To reduce gene flow into natural populations by cultured stocks, alternative strategies for aquaculture, including off-the-coast, offshore, or deeper offshore facilities, should be implemented to protect natural populations from cultured stocks [62]. Finally, it would be necessary not to pollute the ecosystem of wild populations, playing a key role in *M. unguiculatus* resources' protection.

## 5. Conclusions

The genetic diversity of *M. unguiculatus* in the coast of China was at a high level, the genetic structure was relatively weak with nonsignificant genetic differentiation. This population genetics of *M. unguiculatus* was important to reveal its situation in natural environment. We believed that the results from this study can accumulate basic genetic data of *M. unguiculatus*.

**Author Contributions:** Methodology, X.W. and Z.F.; software, Z.F. and J.L.; data analysis and sample processing, Z.F.; writing—original draft preparation, X.W. and Z.F.; writing—review and editing, B.G., J.L. and Y.Y.; funding acquisition, Y.Y. All authors have read and agreed to the published version of the manuscript.

**Funding:** This research was funded by the National Natural Science Foundation of China (42107301), the Fundamental Research Funds for Zhejiang Provincial Universities and Research Institutes (2021J005), and the Project of the Bureau of Science and Technology of Zhoushan (2020C21026).

**Institutional Review Board Statement:** All animal experiments were conducted under the guidance of and approved by the Animal Research and Ethics Committee of Zhejiang Ocean University. Approval code: ZJOU2022078.

**Data Availability Statement:** Data will be provided by the corresponding author upon reasonable request.

**Conflicts of Interest:** The authors declare no conflict of interest.

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
