# Peer review of "Genetic Structure and Phylogeography of Commercial Mytilus unguiculatus in China Based on Mitochondrial COI and Cytb Sequences"

_fishes, doi:10.3390/fishes8020089_

Round 1

Reviewer 1 Report

Please see file.

Thank you.

Author Response

General comments:

This manuscript describes that phylogeography and Genetic Structure of Mytilus unguiculatus Based on Mitochondrial COI and Cytb Sequences. The data is well analyzed, the results are scientifically interesting and well-written. This manuscript is worth publishing in 'fishes'. However, I think that there are some editorial errors that should be corrected.

Answer: Thank you for your comments on the manuscript. Efforts have been made to clarify the following points and correct the mistakes. The modifications based on the suggestions are explained in further detail below. We have directly revised some minor changes. The corresponding revised manuscript and annotated version with changes highlighted by blue color are submitted.

Major Points

There is no mention of FST analysis in Genetic data analysis.

Please describe the FST analysis.

Answer: the FST analysis has been added.

Table3

Have you corrected for multiple comparisons?

Please describe the corrections for multiple comparisons in Genetic data analysis.

Answer: Yes. P-values adjusted for multiple comparisons using Bonferroni correction = 0.05/21.

Page2

Fig.1

I think that the texts are small, please make them bigger in Fig1.

Answer: the graph has been modified.

Page4

Figure1→Figure2

Answer: this has been modified.

References

I think that there are many editorial errors in references.

Some are listed below. Please correct references according to‘Instructions for Authors and latest articles.

Answer: All editorial errors have been modified.

L301

Mytilus coruscus→Mytilus coruscus

Answer: this has been modified.  

L302

  1. De, L.M., Salen P., Monjaud I., Delaye, J.→2. De, L.M.; Salen P.; Monjaud I.; Delaye, J.

Answer: this has been modified.

L310

Trypanosoma cruzi→ Trypanosoma cruzi

Acta Trop→Acta Trop.

Answer: these have been modified.

L328

  1. Swindell, S.R.; Plasterer, T.N.;→ 15. Swindell, S.R.; Plasterer, T.N.

Answer: We had revised this.

L349

  1. Pritchard, J.K.; Wen, X.; Falush, D.;→27. Pritchard, J.K.; Wen, X.; Falush, D.

Answer: We had revised this.

L368-370

Please see L188. Please confirm number 37 and 38.

Answer: this has been modified.

L370

  1. Yang, Z.X.; Mao, Y.L.; Song, N.; Gao, T.X.; Cai, H.C.; Zhang, Z.H.;→38. Yang, Z.X.; Mao, Y.L.; Song, N.; Gao, T.X.; Cai, H.C.; Zhang, Z.H.

Answer: this has been modified.

L386

  1. Shanks, A.L.; Grantham, B.A.; Carr, M.H. (2003) →46. Shanks, A.L.; Grantham, B.A.; Carr, M.H.

Answer: this has been modified.

Reviewer 2 Report

In this research, the authors used mitochondrial DNA markers (COI and Cytb sequences) to evaluate the genetic diversity, structure, population dynamics and phylogeography of the commercial bivalve Mytilus unguiculatus within China. The methodology used was adequate regarding the objectives of this research, and the Discussion would be coherent with the results obtained. Nevertheless, I have found several typos and some critical issues which should be addressed. The pending issues that need to be addressed before the manuscript can be considered for publication are listed in the attacched pdf in two sections: (1) Content issues and (2) Formatting issues.

Author Response

Reviewer 2:

In this research, the authors used mitochondrial DNA markers (COI and Cytb sequences) to evaluate the genetic diversity, structure, population dynamics and phylogeography of the commercial bivalve Mytilus unguiculatus within China. The methodology used wasadequate regarding the objectives of this research, and the Discussion would be coherent with the results obtained. Nevertheless, I have found several typos and some critical issues which should be addressed. The pending issues that need to be addressed before the manuscript can be considered for publication are listed below in two sections: (1) Content issues and (2) Formatting issues.

Answer: Thank you for your comments on the manuscript. Efforts have been made to clarify the following points and correct the mistakes. The modifications based on the suggestions are explained in further detail below. The corresponding revised manuscript and annotated version with changes highlighted by blue color are submitted.

(1) Content issues

Title

—Lines 2-3: The title could be more informative. Maybe it could be better to do something like this:

“Genetic Structure and Phylogeography of commercial Mytilus unguiculatus in China

Based on Mitochondrial COI and Cytb Sequences”.

Answer: this has been modified.

Abstract

—Line 17: I recommend changing “obvious” to “clear”.

Answer: this has been modified.

—Line 19: Maybe I would change “each population” to “in samples”.

Answer: this has been modified.

Introduction

—Lines 32-37: I have missed some further information in these lines. For instance, the global/country capture production in tonnes of live weight (between brackets in any sentence) and maybe its comparison with another species of the same family using the FAO database (see Fig.1). Furthermore, there is as well important mytiliculture production, at least in Korea (https://www.fao.org/fishery/statisticsquery/en/aquaculture). In this country in 2020, myticulture production was remarkably higher than capture.

Figure 1. Screenshot from FAO database for Global capture production Quantity

(https://www.fao.org/fishery/statistics-query/en/capture/capture_quantity)

Answer: this information has been added. According to Food and Agriculture Organization statistics (FAO, 2020), global capture production of M. unguiculatus was rapidly expanding during the 2000 s, with a peak of 3399 tonnes in 2009, but then followed by a sustained decline. In 2020 the annual global capture production of M. unguiculatus was about 633 tonnes (FAO, 2020). However, the global capture production another popular mussel species Perna viridis was about 12000 tonnes (FAO, 2020).

—Line 41: “its genetic information is still ill-formed”. When a bibliographic search is done for Mytilus unguiculatus or unaccepted synonyms as Mytilus coruscus (see the World Register of Marine Species database, WORMS; https://www.marinespecies.org/aphia.php?p=taxdetails&id=506159; Fig.2) there are not many genetic publications about this species. Nevertheless, in the last years, two genome assemblies were published (Table 1), which must be considered and at least mentioned in the main text. This is mandatory for me. Additionally, it would be beneficial for the readers to read an explanatory sentence about how this species has two scientific names used many times (a bit confusing)

Figure 2. Screenshot from WORMS database.

Table 1. Summary of Mytilus coruscus (i.e., Mytilus unguiculatus) published assemblies.

They were related to the available assemblies of this species. Why was a genomic approach using other techniques not employed? For instance, RADseq or Whole-genome resequencing to obtain thousands or millions of nuclear SNPs (Single Nucleotide Polymorphisms) along the genome. These examples would be more expensive, but the latter could be argued in the Introduction and suggested in the Discussion or Conclusion sections as possible next steps with new samples. What was your opinion about this? More information must be included about the genetic/genomic research context of this species (some sentences in the Introduction section).

Answer: Although in the last years, studies on the Whole-Genome Sequencing, Hybrid Assembly and Chromosome-level genome assembly of M. unguiculatus have been reported, the mussel samples they collected were all from the Shengsi Islands in Zhejiang province. Our research scope is larger, covering almost all the mussel producing areas in China. It can well reflect the current status of genetic resources of mussels in coastal areas of China.

The scientific names of the hard-shelled mussel include Mytilus coruscus and Mytilus unguiculatus. “Mytilus coruscus is only widely used in China and is not accepted in the World Register of Marine Species (WoRMS; http://www.marinespecies.org.). According to the rules and priorities of the species double name method, the recognition of “Mytilus unguiculatus” is relatively high in the world academic circles. It can be retrieved in WoRMS. Therefore, this study uses “Mytilus unguiculatus” as the scientific name of this mussels.

Due to the project, we did not use other techniques for analysis. Next, we will use RADseq to study the population genetics of Mytilus unguiculatus.

—Line 44: Please change “my country” to “this country” or “China”.

Answer: “my country” has been changed as “this country”.

—Line 51: I would recommend changing “molecular clock theory” to “implementing the molecular clock theory”.

Answer: “molecular clock theory” has been changed as “implementing the molecular clock theory”.

—Line 64: I would recommend changing “genetic protection units” to “management units (MUs)”.

Answer: “genetic protection units” has been changed as“management units (MUs)”.

Material and Methods

—Line 73: The scale bar of the map (Figure 1) must be wrong. Between Quingdao and Zhoushan, there is 900 kilometres approximately, but according to the map’s scale bar, less than 100 kilometres. Please review it.

Answer: the graph has been modified.

—Line 73: There are different abbreviations in the map not explained in the caption (e.g., BS, JS). Please add them.

Answer: Abbreviations in the map have been explained in the caption.

—Line 74: In the caption of Figure 1, there are “sampling sites”, but immediately in the header of Table 1 (line 74), there are “populations”. I strongly recommend being more homogeneous in the main text. I think it would be adequate in the Material and Methods section to use “sampling locations” or “locations” in all cases. After, in the results, we could speak about one or various populations. With all, taking into account the almost no genetic differentiation found between sampling locations (Results section) from a genetic point of view would be more adequate to use “population”, not “populations”.

Answer: these have been modified.

—Line 77: The software is “Primer premier”. Please review it.

Answer: this has been modified.

—Line 96: Please add the reference of BLAST tools (NCBI). When used, the citation

appears with the results (example in Fig.3). Which database was used for the BLAST

analysis? Nucleotide collection, reference genome of the species? You could have used as Organism database to blast “Mytilus coruscus (taxid:42192)”. It is important to clarify it.

Figure 3. Screenshot from NCBI BLAST tool.

Answer: the reference of BLAST tools has been added. Our approach is consistent with your view.

—Lines 101-102: I would modify “obtained the classic genetic parameters, such as

number of haplotype (n), haplotype diversity (h) and nucleotide diversity (π)” to “usual genetic parameters, such as the number of haplotypes (n), haplotype diversity (h) and nucleotide diversity (π) were obtained”.

Answer: this has been modified.

—Line 101: Please change “Seven populations” to “Sampling locations”.

Answer: “Seven populations” has been changed as “Sampling locations”.

—Lines 112-113: The number of K tested a priori were from 1 to 8, being 8 the number of sampling locations +1. I would indicate this in the main text. Please change (if you agree) “(K set 1 to 8,” to “(K set from 1 to 8, i.e., N sampling locations +1,”.

Answer: this has been deleted.

—Lines 113-116: What K estimator was used from STRUCTURE HARVESTER? L(K), ΔK? This must be clear in Material and Methods. In line 163 (caption of Figure 4; Results section) is mentioned ΔK. This K estimator has a reference (Evanno et al. 2005)3, but it is not included. Furthermore, ΔK can never achieve as most suitable K = 1. The minimum ΔK is always 2. This limitation must be considered. Were considered more K estimators? Please, explain it and clarify it in the main text.

Additionally, there are more web resources to perform this analysis. I recommend StructureSelector as well: https://lmme.ac.cn/StructureSelector/.

Answer: DNA sequences violate the assumptions of Bayesian clustering models available in STRUCTURE, it is not a good option to run STRUCTURE with multiple DNA sequence data using haplotype information. So we remove the STRUCTURE analysis part of this MS.

Results

—Lines 127-128: “With more than 54 individuals, MC-COâ…  -Hap_02 was the central

haplotype, accounting for 28.6%.”. This 28.6% would correspond to 54.0.54 individuals. Then, I would rewrite the beginning of the previous sentence: “With 54 individuals, MCCOI-Hap 02 was…”.

Answer: The sentence has been rewritten.

—Line 131: Please change “In each population” to “In each sample” or “In each

location”. For instance, in line 139, “seven samples” was written, which is more

appropriate considering the results obtained with no genetic differentiation among

sampling locations.

Answer: “In each population” has been changed as “In each sample” .

—Line 148: Please change “among groups within samples” to “among samples within groups”. Please review it.

Answer: “among groups within samples” has been changed as “among samples within groups”.

—Line 155: Was there Bonferroni or another correction of p-values?

Answer: p-values adjusted for multiple comparisons using Bonferroni correction = 0.05/21.

—Line 157: If well, AMOVA analysis can run with one group with uniquely one sample, it would be better to have more sampling locations for group 1. What is your opinion about this? Were all p-values significant (Table 4)?

Answer: the few sampling locations for group 1 are because we found only one sampling point for Mytilus unguiculatus around the Yellow Sea. Most samples from the sampling sites are contaminated by the damaged environment and farming industries. Therefore, we were unable to perform the sampling. P-values adjusted for multiple comparisons using Bonferroni correction = 0.05/21. All p-values were not significant.

—Line 163 (Figure 4): For me, more information in this Figure would be mandatory. Two plots with the mean estimated log-normal (Ln) probability of the data about the simulated number of clusters K (see STRUCTURE manual) should be included, maybe instead of ΔK plots. As was mentioned previously, ΔK ≥ 2. A plot could also be included with the average individual assignment probability (y-axis) of individuals for K = 8 (the max. tested). According to this plot, are there no values for ΔK = 7? With 8 Ks tested, there will be values for ΔK = 7. Finally, I would add “K” as the label for the x-axis (Fig.4)

Figure 4. Screenshot from the manuscript, with modifications made by the reviewer.

Answer: DNA sequences violate the assumptions of Bayesian clustering models available in STRUCTURE, it is not a good option to run STRUCTURE with multiple DNA sequence data using haplotype information. So we remove the STRUCTURE analysis part of this MS.

—Line 167: I would recommend changing “obviously populations events” to “clear demographic events”.

Answer: “obviously populations events” has been changed as “clear demographic events”.

—Line 168: “However, it was lacked the fossil tag of M. unguiculatus.” I did not understand well this sentence here.

Answer: the sentence has been deleted.

—Line 172: I think it would be “effective population size” instead of “effect sample size”; the same for the y-axis of Figure 5.

Answer: these have been modified.  

Discussion

—Lines 192-194: This statement must be softened. Maybe something like that (if you agree): “The high genetic diversity of mussels would indicate that M. unguiculatus resources in Chinese coastal areas were protected in the short term and could have good development prospects.”

Answer: The sentence has been rewritten.

—Line 200: I would add “mostly” in this sentence; “divergence among samples of M. unguiculatus was mostly not significant. There were some significant pairwise FST (table 3).

Answer: “mostly” has been added.

—Line 202: “inter-samples” or “within samples”? According to Table 4, it would be the latter. Please review it.

Answer: this has been modified.

—Line 271: “rare gene” or “rare variants”?

Answer: this has been modified.

—Line 283: “encourage them to isolate the breeding sites and natural environment”. But how? Give me some examples, please. It could be added at the end of this sentence in the main text. Are there many breeding sites in China? Maybe near sampling locations? What was the production? Since when? This information should be included in this subsection (more context).

Answer: Mytilus unguiculatus mainly anchor in the shallow sea area below the low tide line. and there were many breeding sites in Zhejiang and Fujian provinces of China. It could be tried Off-the-coast aquaculture, Offshore aquaculture or Deeper offshore aquaculture to separate the living areas of natural populations from cultured populations. In this way, it would not pollute the water quality resources of natural populations and play a role in protecting wild Mytilus unguiculatus resources. Since 2017, China has developed and built more than 10 kinds of profound Marine aquaculture equipment with complex structures. In June 2021, the equipment located in Qingdao, Shandong province successfully realized the first batch of Chinese deep-sea Salmo salar harvest.

—Lines 284-285: “reduce the genetic exchange between cultured and natural samples”. Then, is there a genetic flow between them? Here lacks information on bibliographic references. “Decrease the contamination of genetic information of natural populations.” were there measures of hybridisation and introgression events? What genetic markers were used in previous research? Again, here lacks more information.

Answer: The existence of genetic flow between natural and cultured populations could be confirmed by Guan et al and Shen et al. And natural populations represent the primary source of genetic variability for aquaculture stocks. However, in artificial breeding, the survival of the fittest was usually carried out, and individuals with higher economic value were selected. In the long run, some populations were gradually eliminated. Once the cultured populations’ degenerates, it would be difficult to find breeding populations in wild populations. In addition, hybridization and genetic introgression also exist among the different species. Shen et al used ISSR markers to study the genetic introgression of Mytilus unguiculatus and Mytilus edulis. They found that hybridization and gene introgression occurred between some Mytilus unguiculatus and Mytilus edulis in Shengsi, Zhejiang. Yang studied the interspecific crossbreeding of Mytilus unguiculatus and Mytilus edulis under laboratory conditions and made the genetic identification of the hybrid progeny by ITS gene. She found that hybridization of Mytilus unguiculatus♀and Mytilus edulis♂was feasible, and 2b-RAD differential SNP cluster analysis found that some Mytilus edulis were significantly more related to Mytilus unguiculatus, a phenomenon most likely caused by hybridization and genetic introgression. Therefore,it was good to reduce the genetic exchange between cultured and natural populations and decrease the contamination of genetic information of natural populations.

(2) Formatting issues.

Below is a non-exhaustive list of typographical or formatting errors.

Introduction

—Line 51: Please change “And compared with” to “Compared with”.

Answer: this has been modified.

—Line 57: Please remove one space. Change “China 's” to “China's”

Answer: this has been modified.

Material and Methods

—Line 74: The first sampling location is in bold. I would explain it in the header of Table 1. I understand that this format is related to Quingdao being the far-away sample (possibly a different population).

Answer: this has been modified.

—Line 88: The substantive must be in the plural form, “each pair of primers was screened in gradient temperature”.

Answer: this has been modified.

—Line 93: Please remove “The” in “The BioEdit v7.25”. Is “BioEdit v7.25” or “BioEdit v7.2.5”? Please review it.7.2.5

Answer: “The” has been deleted. “BioEdit v7.25” has been changed as“BioEdit v7.2.5”.

—Line 93: Please change “utilised” to “used”. The same for line 98.

Answer: these have been modified.

—Line 93: I would remove “software” or write it after “Seqman v8.1”

Answer: “software” has been deleted.

—Line 99: Please change “preformed” to “performed”.

Answer: “preformed” has been changed as “performed”.

—Line 113: Please change “mcmc” to “MCMC”.

Answer: “mcmc” has been changed as “MCMC”.

—Line 116: Please add a space “programBEAST” that must be separated.

Answer: the space has been added.

Results

—Line 139: This is Figure 2, not Figure 1. Please, review it.

Answer: this has been modified.

Discussion

—Line 188: “Yang et al [37]”, nevertheless this article is the 38th reference (line 372,

Reference section).

Answer: this has been modified.

References

—Line 303: “Mytilus coruscus” must be in italics.

Answer: this has been modified.

—Line 312: “Trypanosoma cruzi” must be in italics.

Answer: this has been modified.

—Line 314: “COI and Cytb” genes must be in italics according to the original title of the reference (Laopichienpong et al., 2016).

Answer: this has been modified.

Reviewer 3 Report

Dear authors,

I had read the manuscript titled "Phylogeography and Genetic Structure of Mytilus unguiculatus Based on Mitochondrial COI and Cytb Sequences" with interest. Understanding the genetic structure of species is necessary for the careful and at the same time effective exploitation of resources, and therefore your work is important. The manuscript is well-written and clear.

In my opinion, it would be useful to carry out some analysis of what is known about the diversity and spatial structure of M. unguiculatus populations, both in terms of genetic markers and morphological ones. What do other authors think about the variability of M. unguiculatus? This will slightly increase the list of references, but will make the study, in my opinion, more interesting.

Please note: in lines 148-150, as I understand it, the designations of genes are mixed up. Overall, I think the manuscript is worthy of publication.

Author Response

Reviewer 3:

Dear authors,

I had read the manuscript titled "Phylogeography and Genetic Structure of Mytilus unguiculatus Based on Mitochondrial COI and Cytb Sequences" with interest. Understanding the genetic structure of species is necessary for the careful and at the same time effective exploitation of resources, and therefore your work is important. The manuscript is well-written and clear.

Answer: Thank you for your comments on the manuscript. Efforts have been made to clarify the following points and correct the mistakes. The modifications based on the suggestions are explained in further detail below. The corresponding revised manuscript and annotated version with changes highlighted by blue color are submitted.

In my opinion, it would be useful to carry out some analysis of what is known about the diversity and spatial structure of M. unguiculatus populations, both in terms of genetic markers and morphological ones. What do other authors think about the variability of M. unguiculatus? This will slightly increase the list of references, but will make the study, in my opinion, more interesting.

Answer: this has been added.

Please note: in lines 148-150, as I understand it, the designations of genes are mixed up. Overall, I think the manuscript is worthy of publication.

Answer: this has been modified.

Round 2

Reviewer 2 Report

The authors have implemented all the suggested changes recommended in the first revision in this new manuscript version, adequately arguing the issues raised. The quality of the manuscript has improved enough. Nevertheless, I observed some typos that must be corrected and some jumbled sentences in the Introduction and the Discussion sections that must be improved. I have attached a pdf file.

Author Response

Reviewer 2:

The authors have implemented all the suggested changes recommended in the first revision in this new manuscript version, adequately arguing the issues raised. As a result, the quality of the manuscript has improved enough. Nevertheless, I observed some typos that must be corrected and some jumbled sentences in the Introduction and the Discussion that must be improved. The pending issues that need to be addressed before the manuscript can be considered for publication are listed below in two sections: (1) Content issues and (2) Formatting issues.

Answer: Thank you for your comments on the manuscript. Efforts have been made to clarify the following points and correct the mistakes. The modifications based on the suggestions are explained in further detail below. We have directly revised some minor changes. The corresponding revised manuscript and annotated version with changes highlighted by blue color are submitted.

(1) Content issues

Introduction

—Lines 32-42: These sentences could be improved for more clarity. This is mandatory for me. For instance, something like this: “The hard-shelled mussel has diff scientific names such as Mytilus coruscus and Mytilus unguiculatus. Although, if well, “Mytilus coruscus” is widely used in the bibliography, this form is not accepted according to the World Register of Marine Species (WoRMS;  http://www.marinespecies.org). Therefore, this study uses “Mytilus unguiculatus” as the scientific name of this mussel. This species inhabits the temperate zone along the coastal water of the Indo-Pacific region. According to Food and Agriculture Organization (FAO), global capture production of M. unguiculatus was rapidly expanding during the 2000 s, with a peak of 3399 tonnes in 2009, but then followed by a sustained decline (633 tonnes in 2020).” I would add after this one sentence some causes of the production decline.

Answer: this has been modified and the sentence has been added.

—Lines 50-54: “Although in the last years, studies on the Whole-Genome Sequencing, Hybrid Assembly and Chromosome-level genome assembly of M. unguiculatus have been reported; the samples they collected were all from the Shengsi Islands in Zhejiang province”. With the “Although” the province used in these studies seems to be a problem when this province is sampled in this study. I would suggest this if you agreed:

“This species is rich in physiological and biochemical functions and disease control.Although in the last years, different genome assemblies of M. unguiculatus have been published [5,6], population genetics/genomics studies are still scarce.”

Answer: the sentence has been rewritten.

Discussion

The last paragraph of the discussion could be clearer. It must be rewritten.

—Lines 192-319: Maybe something like this.

“In present studies, the decline of M. unguiculatus resources was reported [52,53]. Therefore, setting up a corresponding genetic protection unit may represent an effective way to protect germplasm genetic resources. It also should limit the physical communication between cultured populations and the wild ones to reduce the possibility of gene flow between them as far as possible. The existence of this gene flow was confirmed by Guan et al. [37] and Shen et al. [54]. Furthermore, these wild populations represent aquaculture stocks' primary source of genetic variability [54]. Hybridisation and genetic introgression also exist among different mussel species (e.g., Mytilus edulis; [55,56]). M. unguiculatus mainly anchor in the shallow sea area below the low tide line [57] with many overlapped breeding sites in the Zhejiang and Fujian provinces of China [58,59]. To reduce this gene flow could be tried off-the-coast aquaculture, offshore aquaculture, or deeper offshore aquaculture to separate the living areas of wild populations from cultured populations [60]. Finally, it would necessary not to pollute the ecosystem of wild populations, playing the latter a key role in M. unguiculatus resources protection.” I would remove the last sentences and their references [61, 62, 63]

Answer: this has been modified.

(2) Formatting issues.

Below is a non-exhaustive list of typographical or formatting errors.

Material and Methods

—Line 121: Please change “testsof ” to “tests of ”

Answer: this has been modified.

—Line 131: Please change “permutation” to “permutations”

Answer: “permutation” has been changed as “permutations”
